# Adaptive Responses of a Peroxidase-like Polyoxometalate-Based Tri-Assembly to Bacterial Microenvironment (BME) Significantly Improved the Anti-Bacterial Effects

**DOI:** 10.3390/ijms24108858

**Published:** 2023-05-16

**Authors:** Chunxia Zhang, Rongrong Liu, Xueping Kong, Hongwei Li, Dahai Yu, Xuexun Fang, Lixin Wu, Yuqing Wu

**Affiliations:** 1State Key Laboratory of Supramolecular Structure and Materials, College of Chemistry, Jilin University, No. 2699 Qianjin Street, Changchun 130012, China; zcx19941002@163.com (C.Z.); kongxp17@mails.jlu.edu.cn (X.K.); lihongwei@jlu.edu.cn (H.L.); wulx@mails.jlu.edu.cn (L.W.); 2Key Laboratory for Molecular Enzymology and Engineering of Ministry of Education, College of Life Science, Jilin University, No. 2699 Qianjin Street, Changchun 130012, China; rongrong20@mails.jlu.edu.cn (R.L.); yudahai@jlu.edu.cn (D.Y.); fangxx@jlu.edu.cn (X.F.); 3Institute of Theoretical Chemistry, College of Chemistry, Jilin University, No. 2 Liutiao Road, Changchun 130023, China

**Keywords:** polyoxometalates (POMs), adaptive response, biogenic amine, antibacterial enhancements, bacterial microenvironment (BME)

## Abstract

The present study presents the tertiary assembly of a POM, peptide, and biogenic amine, which is a concept to construct new hybrid bio-inorganic materials for antibacterial applications and will help to promote the development of antivirus agents in the future. To achieve this, a Eu-containing polyoxometalate (EuW_10_) was first co-assembled with a biogenic amine of spermine (Spm), which improved both the luminescence and antibacterial effect of EuW_10_. Further introduction of a basic peptide from HPV E6, GL-22, induced more extensive enhancements, both of them being attributed to the cooperation and synergistic effects between the constituents, particularly the adaptive responses of assembly to the bacterial microenvironment (BME). Further intrinsic mechanism investigations revealed in detail that the encapsulation of EuW_10_ in Spm and further GL-22 enhanced the uptake abilities of EuW_10_ in bacteria, which further improved the ROS generation in BME via the abundant H_2_O_2_ involved there and significantly promoted the antibacterial effects.

## 1. Introduction

Bacterial infections have always been a severe threat to the global economy and human health [1]. With the emergence of the COVID-19 global pandemic, significant challenges have come to the fore to urgently fight off viral and bacterial damage. Although antibiotics have significantly contributed to efficiently inhibiting or killing bacteria by inhibiting protein synthesis or preventing DNA replication, the emergence of drug-resistant strains in patients has relit the flames of this struggle [2]. With high efficiency and safety as antibiotic alternatives, developing a new class of antibacterial materials has become practicable. Recently, newly developed antibacterial materials have been widely applied to medical treatment, food packaging, water treatment, and biological medicine [3,4,5]. With the development of nanotechnologies, significant efforts have been devoted to discovering new antibacterial nanocomposites, including those constructed between polyoxometalates (POMs) and cationic polymers, proteins, specific peptides, or other biomolecules [6,7].

Polyoxometalates (POMs) are negatively charged nanoclusters composed of transition metals, mostly Mo, W, and V with oxygen [8]. The broad structural diversity and outstanding physicochemical properties resulted in their wide application in material science [9], photochemistry and electrochemistry [10], protein crystallography [11], catalysis [12], and particularly medicine [13]. The easily modulated interactions between biological targets and POMs enhanced their activities beneficially on the respective biological system [14]. This led to systematic studies of their antibiotic effects [15] beyond diabetes as well as [16] antitumor [17] and antiviral [18] effects. There is no doubt that the specific electrostatic interactions between POMs and biomolecules contributes greatly to their activities, where the negatively charged metal clusters are essentially within or at positively charged regions of the proteins/peptides [19,20]. A review summarized the structures, antibiotic effects, and future perspectives of POMs on the antibacterial activity [21], which illustrates the most prominent POM structures used for antibacterial studies. Beyond the activity of purely inorganic POMs, the antimicrobial progress of POM-based nanocomposites highlights the recent developments in this field via supramolecular assembly with peptides [22] and proteins [23]. However, its antibacterial properties are unclear, as is the ROS generation. Therefore, it is important to study the ROS release of EuW_10_ and the assembly with Spm and GL-22 in different sequences, which evidently contribute to antibacterial applications.

Spermine (Spm) is a small polycation derived from amino acids through spermine synthase or spermine oxidase, which is a deficiency of polyamines that gives rise to neurological diseases such as epilepsy, while GL-22 is a cationic peptide from HPV16 early oncoprotein (E6). In the present study, we construct a tertiary assembly using a Weakley-type POM, Na_9_[EuW_10_O_36_]·32H_2_O (EuW_10_) [24], a biogenic amine of spermine (Spm), and arginine-rich peptides (GL-22) from HPV E6 early oncoprotein. What is most interesting is that with positive charges at the surfaces, Spm and GL-22 did not form competitive binding with EuW_10_ rather than a synergistic process between them, which led to a step-up fluorescence enhancement and significantly improved the antibacterial activity in comparison with EuW_10_, and even two binary assemblies among them (Figure 1). Moreover, the multi-stage binding model between them revealed that the spherical tri-assembly was driven by the cooperative electrostatic interaction between the constituents. The luminescence enhancement of EuW_10_ was essentially attributed to the assembly with Spm and GL-22 or the surface-water shielding, where the ROS is generated during incubation with Escherichia coli (*E. coli*). Therefore, the EuW_10_/Spm/GL-22 assembly is regarded as an ROS factory that extensively impacts bactericidal activities. The well-defined nanoparticle from EuW_10_, Spm and peptide GL-22 is illustrated as a model, which provided a way to fabricate multi-stage bioinorganic materials with solid fluorescence and expand the antibacterial mechanism of POMs through the emergence of ROS. The present study supplies a tertiary assembly between POM, peptide, and biogenic amine, which is a concept to construct unique hybrid bio-inorganic materials for antibacterial applications and will be helpful in the promotion of the development of antiviral agents in the future.

## 2. Results

### 2.1. Construction and Characterizations of the Tertiary Assembly of EuW_10_/Spm/GL-22

The construction of the tri-assembly between EuW_10_, polyamine, and peptide was monitored using the fluorescence titration spectra (Figure 1). The luminescence of EuW_10_ enhanced upon Spm being added into the solution containing 50 μM EuW_10_, which reached a maximum at 50 μM Spm (Appendix A–C). Such enhancements were attributed to the surrounding water being expelled from the coordination of Eu(III)/water [25,26]. A 21.92-fold emission enhancement was achieved for EuW_10_/Spm over EuW_10_ (Figure 1A), confirming strong binding between EuW_10_ and Spm (Appendix A). Further titration of GL-22 into the solution containing the EuW_10_/Spm assembly step-up enhanced the luminescence, which reached a maximum at 35.0 μM GL-22 (Figure 1B) but reduced along with more GL-22 joining (Figure 1C), which was attributed to a synergistic rather than the competitive binding between them [26]. The plot in Figure 1D summarizes the intensity changes in EuW_10_ at 591 nm upon the introduction of Spm and the following GL-22 peptide, which suggests that a more hydrophobic microenvironment was produced for Eu(III) by forming a tertiary assembly of EuW_10_/Spm/GL-22. In addition, the Energy dispersive spectrum of EuW_10_/Spm/GL-22 (50 μM/50 μM/35 μM) is characterized in Appendix A.

As revealed by DLS (Figure 2), the bi-assembly of EuW_10_/Spm (50 μM/50 μM) contributed an averaged particle size of 37.8 nm in the buffer. As assayed by DLS (Appendix A), after a 30-day incubation in the buffer solution, little change in the particle size during assembly was observed, which validates its excellent stability. Further addition of GL-22 (5 μM) resulted in a particle size of 50.7 nm. Then, along with a titration of more GL-22, the particle size increased to 165 nm at 35.0 μM GL-22 (Figure 2A). Finally, at 70 μM GL-22, it increased to around 295 nm. Therefore, the particle size of the assembly successively increased along with the titration of GL-22 (Figure 2B). Moreover, the surface potential of EuW_10_ upon assembling with Spm and GL-22 was evaluated by a Zeta-potential assay (Figure 2C). It approached equilibrium with the introduction of Spm (Figure 2D) and then GL-22, which confirmed the electrostatic-driving interaction between EuW_10_ and two positive constituents.

The tertiary assembly with the reverse adding sequence of Spm and GL-22 was also explored. As a result, the fluorescence of EuW_10_ was step-up amplified with a 38.2-fold enhancement at 591 nm in total. Moreover, the corresponding particle size increased via a different kinetic where a quick increase at a low concentration of Spm was revealed [26], confirming the construction of the EuW_10_/GL-22/Spm assembly differently.

The morphology changes in EuW_10_/Spm upon GL-22 titration were then monitored using transmission electron microscopy (TEM; Figure 3). When 5 μM of GL-22 was presented in the solution containing EuW_10_/Spm assembly, the particles were mono-dispersed with a mean diameter of 51.0 nm (Figure 3A). Low concentrations of GL-22 did not induce bigger spheres, but the cross-linked EuW_10_/Spm was observed (Figure 3B,C). Upon the successive addition of more GL-22, the cross-link network thickens and some aggregates start to enlarge to bigger spherical particles of between 100 and 300 nm (Figure 3D–F). Therefore, the results demonstrate that EuW_10_/Spm and GL-22 first assembled into the cross-link, producing larger particles along with more GL-22 joining. Finally, at 70 μM of GL-22, the nanospheres aggregate into larger particles with diameters of between 200 and 300 nm.

Therefore, the interaction between EuW_10_/Spm and GL-22 resulted in the tri-assembly of EuW_10_/Spm/GL-22, which was further validated by DLS, the Zeta-potential, and TEM imaging. Meanwhile, the cross-link binding between EuW_10_/Spm and GL-22 changed the microenvironment and further limited the vibration and rotation of the surface molecules, which enhanced the emission by reducing the non-radiative decay of Eu(III).

### 2.2. Enhanced Antibacterial Effects of the Bi- and Tri-Assembly

The assembly with positive Spm and GL-22 changed the surface charges of EuW_10_, which may be beneficial for the trans-membrane ability and encourage us to expand the exploration of their antibacterial capacities. The tri-assembly of EuW_10_/Spm/GL-22 was first tried on Gram-negative Escherichia coli (*E. coli*) based on a standard plate method to evaluate the bacterial viability [27]. Meanwhile, the individual EuW_10_, Spm, GL-22, and their binary assemblies were tested, respectively, for comparison (Figure 4A). A significant reduction in the bacterial colony was shown after incubation with the binary and tri-assemblies. According to the Plaque count method (Figure 4B), the individual Spm and GL-22 in the plate show almost no inhibition of bacterium growth compared to the blank, indicating that neither of them has an antibacterial effect on *E. coli*. On the other hand, EuW_10_ demonstrated the inhibition of *E. coli*. growth with a 65.22% viability left, which is consistent with a previous report [6] and validates the current approach of using an antimicrobial assay. The antibacterial activity of 29 different POMs against Moraxella catarrhalis was reported, which is essentially dependent on the particle size, shape, composition, and surface net charges; finally, the minimum inhibitory concentration (MIC) was achieved in a range of 1–256 μg·mL^−1^ for them [28]. Compared with these, a moderate antibacterial capacity (MIC = 167.55 μg·mL^−1^) was achieved for EuW_10_, which needs improvement via assembly with biomolecules as biogenic amines and/or peptides.

All the results were statistically analyzed using the analysis of variance (ANOVA) methods. As a result, the bi-assembly with Spm and GL-22 decreased the viability of *E. coli* to 31.44% and 23.08%, respectively. More interestingly, the tri-assembly of EuW_10_/Spm/GL-22 resulted in the thorough sterilization of *E. coli*. Finally, an MIC of 41.88 μg·mL^−1^ was achieved for tri-assembly against *E. coli*, which was significantly promoted compared to EuW_10_ and the bi-assemblies. In addition, among the reported antibacterial materials (Appendix A) [29,30,31,32,33,34], the efficiency of the constructed tri-assembly is comparable (*p* < 0.001).

In parallel, the survival rate of another tri-assembly, EuW_10_/GL-22/Spm, on *E. coli* was also tested, which showed a slightly higher antibacterial effect of 8.03% more than EuW_10_/Spm/GL-22 compared to being constructed from a different adding sequence of two constituents (Figure 4). Therefore, significantly enhanced antibacterial activity was achieved for both tri-assemblies.

In addition, Gram-positive Staphylococcus aureus (*S. aureus*) was employed to extend the applications of the developed assemblies on disinfection (Figure 5A), using the same method created for *E. coli*. Compared to the blank, 6.25 μM GL-22 showed a slight inhibition of *S. aureus* growth with a viability of 84.71%, while 12.5 μM Spm displayed apparent inhibition of it with only a 2.34% viability left (Figure 5B). Several earlier studies showed that exogenous Spm became toxic to bacteria with consistent accumulation; its ability to inhibit bacterial growth is high [35]. Moreover, the joining of GL-22 into the EuW_10_/Spm further enhanced the antibacterial capacity, where only 0.25% viability was maintained. Similarly, the survival rate of EuW_10_/GL-22/Spm against *S. aureus* dropped to 0.12% in identical conditions. Moreover, it was reported [36] that conjugation of Spm with stapled peptide further enhanced the cellular uptake of the bioactive peptides, which supported well the improved antibacterial effect of tri-assemblies. Moreover, the concentration of each constituent was lowered to one-fifth to evaluate the actual anti-bacterial effects on *S. aureus* more clearly (Appendix A). The inhibitory ability of EuW_10_/Spm/GL-22 and EuW_10_/GL-22/Spm against *S. aureus* was evaluated to be 6.29% and 11.10%, respectively. Finally, an MIC of 20.94 μg·mL^−1^ was evaluated for the tri-assembly of EuW_10_/Spm/GL-22 against *S. aureus*, which was significantly promoted in comparison to that of EuW_10_ (251.38 μg·mL^−1^). As feeble antibacterial effects of EuW_10_ and GL-22 were shown on *S. aureus*, a more significantly improved disinfection of EuW_10_/Spm/GL-22 compared to EuW_10_ and EuW_10_/GL-22 was achieved because of the synergistic effects between them.

Recent reports demonstrated that the antibacterial efficacy could be improved significantly when the cationic peptides were self-assembled with POMs [22,37,38]. Moreover, the cysteine group of the peptide can dramatically enhance the binding affinity through simultaneous interactions with bacterial membranes [39]. The action of surface peptides was proposed to follow the consecutive steps, electrostatic binding, and accumulation on the membrane surface, followed by the insertion into the cell membrane, which finally induced cell lysis [40]. Moreover, Spm assembly may accelerate these processes [35], which contributed to the enhanced antibacterial efficacy of the tri-assembly.

The antibacterial activity of inorganic POMs can be divided into two primary parts, direct and synergistic effects [21]. As EuW_10_ did not exhibit strong antibacterial activity at the pharmacological concentrations, the contribution by Spm and GL-22 on EuW_10_ should be crucial for such improvements. Under the identical condition, EuW_10_/Spm/GL-22 completely suppressed the bacterium growth (Figure 4 and Figure 5), which relates to the later joining of Spm and GL-22 on the particle surface. Moreover, based on the emission intensity, both the tri-assemblies were stable enough for 3-month preservation in buffer solution, and their antibacterial properties were conserved well. Regarding the similar constituents in different assemblies, the results demonstrate that the two tri-assemblies were more powerful than the binary systems. Moreover, *S. aureus* was relatively more sensitive than *E. coli* to a sample at the same concentration. This may be because Gram-negative *E. coli* has an extra outer membrane barrier to be surmounted compared to Gram-positive *S. aureus* [41].

### 2.3. Dose-Dependent Antibacterial Effects of the Constituent in Tri-Assemblies

The dose dependence of the assembly-enhanced antibacterial effects was evaluated on *E. coli* for the assembly of EuW_10_/Spm/GL-22 in the presence of different amounts of GL-22 (5.0–70.0 μM), respectively (Figure 6A). As indicated, the antibacterial effects of the tri-assembly firmly depend on the constitutional ratios and the incubation time. The prolonged incubation time improved the disinfection obviously as 9 h leads to the most powerful effect among all conditions. Of note, however, the disinfection effects of the tri-assembly did not truly positively correlate with the amounts of GL-22 as the viability bounced at 100 μM GL-22. Specifically, taking a 3 h incubation as an example, EuW_10_ maintained 90.26% viability of *E. coli* whereas EuW_10_/Spm maintained only 83.09% (Appendix A). Sustainably, the antibacterial effect was enhanced via the joining of GL-22 in the tri-assembly. Specifically, it showed a clear dose-dependent manner of GL-22 concentrations and dropped to 40.73% at 35 μM, although it bounced again along with more GL-22 joining. Therefore, the best antibacterial effect of EuW_10_/Spm/GL-22 on *E. coli* was evaluated at EuW_10_/Spm (50 μM/50 μM) in the presence of 35 μM GL-22, where the most vigorous emission (Figure 1) and moderate particle size of 142 nm were achieved (Figure 3D), although their intrinsic relationships are still unclear.

Of note, similar results were also achieved regarding *E. coli* for the tri-assembly of EuW_10_/GL-22/Spm by changing the join sequence (Figure 6B). For example, the bi-assembly of EuW_10_/GL-22 maintained an 80.88% viability of *E. coli* after 3 h incubation. Furthermore, the introduction of different amounts of Spm into the EuW_10_/GL-22 system improved the disinfection effects remarkably. At 25 μM Spm, the survival rate induced by the tri-assembly decreased to 33.16%, illustrating a noticeable improvement over EuW_10_/GL-22. Consequently, the antibacterial effects of EuW_10_/Spm/GL-22 were optimized while the best result was achieved at EuW_10_/GL-22 (50 μM/25 μM) in the presence of 25 μM Spm. Taken together, it substantiates that both tri-assemblies showed robust and effective antibacterial effects, which are dose- and time-dependent.

## 3. Discussion

### 3.1. Intrinsic Mechanism behind the Enhanced Antibacterial Effects of Tri-Assembly

#### 3.1.1. The Peroxidase-like Activity of EuW_10_

POMs are treated as electron reservoirs because they have a strong capacity to bear and release electrons, demonstrating their high redox nature [42]. At the highest oxidation states, the transition metals in POM show peroxidase-like activity by changing the metal state in the structure. For example, Zhang et al. [43] reported that tungsten (W)-based POM clusters (GdW_10_O_36_) have an excellent ability to produce ·OH, where a reductive chemical conversion of W(VI) to W(V) was essential. Herein, the peroxidase-like activity of EuW_10_ and assemblies were measured using TMB as a substrate (Appendix A). Further experiments show that the assemblies significantly promote TMB oxidation. For example, in the presence of EuW_10_, the absorbance of oxTMB changed weakly (Appendix A); however, the EuW_10_/Spm increased the characteristic absorption of A_652_ by 14.6-fold within 5 min (Appendix A). In identical conditions, the oxidation rate of TMB is significantly increased by EuW_10_/Spm/GL-22 as A_652_ increases faster than EuW_10_ and EuW_10_/Spm (Appendix A). Moreover, the peroxidase-like activity of EuW_10_ will be assessed using a DCFH-DA method [44], where the produced ROS will oxidize the non-fluorescent DCFH to the emitted DCF as an indicator. For that, a small amount of H_2_O_2_ was added to the solution containing EuW_10_ and DCFH-DA. Of note, the EuW_10_ produced large amounts of ROS (Appendix A) over the blank (Appendix A). Moreover, in comparison to the condition without H_2_O_2_ participation (Appendix A), the critical impact of H_2_O_2_ on ROS generation was revealed for EuW_10_. Therefore, the EuW_10_-induced ROS generation was directly related to its peroxidase-like activity; the H_2_O_2_ was activated to ·OH by capturing electrons from the LUMO of tungsten.

#### 3.1.2. The Enhanced ROS Generation of Tri-Assembly over EuW_10_ in *E. coli*

After incubation with *E. coli*, EuW_10_ and GL-22 alone produced small amounts of ROS, which may be attributed to the peroxidase-like activity of POMs [43] and a direct free-radical scavenging effect of peptides; [45] however, no ROS generation was shown for Spm compared to the blank (Appendix A), illustrating the weak ability of the individual EuW_10_ and GL-22 to produce ROS in *E. coli*. In the presence of bi- or tri-assemblies, the ROS generation far exceeds the two or three constituents together. EuW_10_/Spm and EuW_10_/GL-22 promoted the ROS generation to 6.0- and 3.55-fold, while the EuW_10_/Spm/GL-22 and EuW_10_/GL-22/Spm achieved 8.77- and 7.77-fold that of EuW_10_ in *E. coli*, respectively (Figure 7). The low level of ROS is beneficial for the cellular life cycle, while abundant ROS will damage the cellular constituents as lipids, proteins, and DNA [46]. Therefore, the improved ROS generation will help to destroy the bacteria as strong oxidants and enhance the sterilization. However, the origination of the enhanced ROS generation by the assemblies in *E. coli* needs to be clarified.

Therefore, we further evaluated the ROS generation by the tri-assemblies in vitro. There is almost no ROS generation for EuW_10_ assemblies in buffers (Appendix A), which should be attributed to the H_2_O_2_ absence. However, in the H_2_O_2_ presence, the assemblies produced remarkable ROS, although it is lower than that of EuW_10_ alone (Appendix A). Compared with the conditions without H_2_O_2_ participation (Appendix A), the impact of H_2_O_2_ on ROS generation was revealed for EuW_10_ and its assemblies (Appendix A). Therefore, the large amounts of ROS generation in the bacteria by EuW_10_ assemblies should be closely related to the involved H_2_O_2_; both the inside and released H_2_O_2_ from the bacteria triggered the ROS yield by the assemblies (Figure 7).

Moreover, the ROS yields of EuW_10_, EuW_10_/Spm, and EuW_10_/Spm/GL-22 in buffers were compared in the presence of H_2_O_2_ (Appendix A). The single EuW_10_ produced abundant ROS in the presence of H_2_O_2_, supported by the peroxidase-like activity of tungsten-containing POM [43]. Therefore, we conclude that the bits of H_2_O_2_ in BME trigger the catalytic efficiency of EuW_10_, resulting in the improvement of ROS, bacterial death, and leakage of internal solutes, which eventually lead to the production of large amounts of ROS and significant antibacterial ability improvement. Such a result exposes the catalytic active facets, and the EuW_10_ acts as an electron acceptor receiving an electron from DA, easily facilitated by the reduced molecular distance under the effect of hydrogen bonding and electrostatic interaction [43]. The W–O bonds acted as active species, interacting with H_2_O_2_, which improved its adsorption on the catalyst surface. Then, H_2_O_2_ was reduced to ·OH by capturing electrons from LUMO of EuW_10_, where the dyes were finally degraded by the ·OH [47]. Therefore, the catalytic efficiency of EuW_10_/Spm/GL-22 synergism enables relatively lower concentrations of H_2_O_2_ to reach the required catalytic sites compared to the individual EuW_10_ and assemblies, which finally results in the death of bacteria.

However, the ROS generation of the binary and tri-assemblies were lower than that of EuW_10_ alone under the same conditions in vitro. Therefore, we speculate that the binding of positively charged Spm and GL-22 at the EuW_10_ surface may somewhat shield the catalytic site and inhibit ROS production. However, in *E. coli*, the assemblies generated more ROS than the EuW_10_. Figure 7B illustrates the different ROS generation between EuW_10_ and the assemblies in *E. coli* and buffers with H_2_O_2_. Moreover, it should be noted that the assemblies indeed promoted the antibacterial effect on *E. coli*. (Figure 6). Therefore, although ROS generation is the leading cause for most antibacterial samples, nevertheless, as it has been reported that the antimicrobial activity of EuW_10_ does not induce large amounts of ROS generation, this suggests that other factors may play vital roles in the antibacterial effect beyond the peroxidase-like activity of samples, which may be related to the size and surface charges of EuW_10_ and its changes via assembly.

### 3.2. Assembly Promoted EuW_10_ Uptake

Compared with EuW_10_, the bi- and tri-assemblies of EuW_10_ produced less ROS in the presence of H_2_O_2_ in vitro (Appendix A), which was attributed to the covered active site through assembly [48]. Therefore, the peroxidase-like activity of EuW_10_ was suppressed after assembling with the positively charged peptide or biogenic Spm. However, the antibacterial ability of the assemblies was indeed improved, there must be other cruxes involved.

In the present study, both the Spm and GL-22 process the amine group and take advantage of surface charges to bind with EuW_10_. We characterized the surface charge changes in EuW_10_ upon assembling with Spm and GL-22 using the Zeta-potential technique (Figure 2C,D). The observed data shift from negative to positive, providing an effective pathway to attach and pass through the bacteria membrane [49]. In addition, it was reported that the materials showed more activity against Gram-negative bacteria than against Gram-positive ones due to their easier internalization [50]. Therefore, the promoted bacteriostatic effects of the EuW_10_ assemblies should also be related to the bacteria internalization.

The positive coating on the EuW_10_ surface improved its uptake to the bacterial membrane; the abundant H_2_O_2_ was catalyzed to produce an abundance of ROS and consequently promoted the antibacterial performance of the assemblies (Appendix A). Constructing the composite material with different constitutions played an essential role in the cooperation of each component to achieve the most synergistic anti-microbial effect, which provided a commendable foundation for the improved antibacterial effect of the EuW_10_ assemblies.

### 3.3. Enhanced Biofilm Elimination of Assemblies

The bacteria biofilm adheres to the surface of material or tissue by packing the extracellular polymeric substances and prevents antibiotic permeation into the bacterial cells [51,52,53]. Consequently, drug resistance to bacteria was produced, and the antibiotic treatment failed. Therefore, as an obstacle, biofilm should be eliminated for antibacterial performance. A staining method using crystal violet was used herein to investigate the biofilm formation and the elimination of individual EuW_10_ or its assemblies (Figure 8A). The intense navy blue shown for the control group, without agent treatment, indicates the biofilm formation for *E. coli* under the current conditions. For the groups following treatment with single EuW_10_, Spm, and GL-22, the slight blue indicates the biofilm formation of 37.7%, 24.9%, and 61.6%, respectively, over the control group. With the introduction of the bi-assemblies of EuW_10_/GL-22 and EuW_10_/Spm, the biofilm formation rates fluctuated to a small extent (38.8% and 31.0%). However, the tri-assemblies of EuW_10_/Spm/GL-22 and EuW_10_/GL-22/Spm have an apparent ability to inhibit the biofilm, and the formation rates of the biofilm were 26.8% and 21.8%, respectively (Figure 8B). In comparison with that of the constituent, there seems to be little improvement in biofilm elimination. Therefore, other factors should be involved for the enhancement of the antibacterial effect by the assembly.

### 3.4. Correlation between Particle Size of the Assembly and Antibacterial Effect

The above discussion suggests that the antibacterial effect may also be influenced by the particle sizes of assemblies. As revealed by the TEM images, at 5 μM GL-22 (Figure 3A), the tri-assembly of EuW_10_/Spm/GL-22 forms part of the mesh complex structure with a size of 50.7 nm; simultaneously, the antibacterial efficiency is 83.09% after 3 h incubation (Figure 6A). At 25 μM GL-22 (Figure 3C), the reticular structure gradually condensed into spheres with particle sizes of more than 100 nm; correspondingly, the bacterial viability dropped to 48.58% in parallel. Moreover, 35 μM GL-22 (Figure 3D) promoted the particles to become more condensed with a sphere diameter of 132 nm, while the bacterial viability dropped to 40.73%. Therefore, a correlation between the disinfection efficiency and particle sizes of the assembly was established. Such a relationship was further validated by the tri-assembly of EuW_10_/Spm/GL-22 (Figure 6B). The highest antibacterial efficiency (33.16%) was achieved at 25 μM Spm for the assembly of EuW_10_/GL-22/Spm after 3 h of incubation, where the spheres showed a mean diameter of ~142 nm. The reports for AgNMs [54] and AgNPs [55] revealed that when the particle sizes were between 100 and 200 nm, they showed excellent antibacterial activity against both *E. coli* and *S. aureus*, which supports the above results.

Meanwhile, along with the incubation time, the viability of the bacteria sustainably decreased, finally reaching 0.51% after 9 h of incubation for EuW_10_/Spm/GL-22, at 35 μM GL-22 (Figure 6A); an antibacterial effect of 0.52% was achieved for EuW_10_/GL-22/Spm in the presence of 25 μM Spm. Among the different assemblies, the ebb of bacteria viability was shown for the relatively uniform spherical morphology with a size of 120~150 nm, along with the incubation time, indicating that a moderated particle size is more conducive to achieving an antibacterial effect.

## 4. Materials and Methods

### 4.1. Chemicals and Materials

Spermine (Spm), 2′,7′-Dichlorodihydrofluorescein diacetate (DCFH-DA) and crystal violet were purchased from Aladdin Chemical Co. Ltd. (Aladdin, Shanghai, China) The peptide of GL-22, with a sequence of GRWTGRCMSCCRSSRTRRETQL from HPV E6 protein, was ordered from Apeptide Co., Ltd. (Apeptide, Shanghai, China). Its purity is higher than 99%, as confirmed by the HPLC performance from the company. Na_9_[EuW_10_O_36_]·32H_2_O, abbreviated as EuW_10_, was synthesized and characterized according to the published procedure [24]. 2-(N-morpholino) ethanesulfonic acid (MES) and sodium hydroxide (NaOH) were purchased from Beijing Chemical Factory (Beijing Chemical Works, Beijing, China). All the chemicals were used as obtained without further treatment. Distilled water (ρ = 18.2 MΩ cm, 25 °C) was obtained from a Millipore Milli-Q water purification system. In addition, the MES-NaOH buffer (10.0 mM, pH = 6.0) was prepared with 10.0 mM MES and NaOH by distilled water. The stock solution of EuW_10_ and Spm was prepared at 2.0 mM in an aqueous solution and placed under dark conditions (4 °C). Then, it was diluted to the required concentrations according to different experimental requirements. Luria Broth (LB) was purchased from Sigma-Aldrich, Wicklow, Ireland. Gram-negative *E. coli* and Gram-positive *S. aureus* were obtained from the Key Laboratory for Molecular Enzymology and Engineering of Ministry of Education, College of Life Science of Jilin University.

### 4.2. Equipment and Characterization

Fluorescence spectra were recorded on a Shimadzu RF-5301PC Fluorescence spectrophotometer (Kyoto, Japan). The excitation lamp was kept on for 0.5 h before the spectra recording to obtain a reliable result. A quartz cuvette (1.0 × 1.0 cm) was used, and 500 or 488 nm (for DCF detection) was fixed as the excitation wavelength for the luminescence record. All the fluorescence measurements were performed at 25 °C in MES-NaOH buffer (10.0 mM, pH = 6.5). Each measurement was repeated three times, and the representative displayed is one of them. The UV–vis absorption spectra were measured using a Shimadzu UV-3600 spectrophotometer (Japan), and all measurements were performed in 10.0 mM MES-NaOH buffer (pH = 6.5) at 1 cm × 1 cm quartz cuvettes. A microplate reader (Bio-RAD iMark) was used to record the absorbance at 595 nm. The optical density (OD) of the sterile LB medium was subtracted from that of the biofilm formed to eliminate any background effects. Each determination was performed in triplicate. A VERTEX 80 V IR (Bruker) spectrometer was used to record the corresponding IR spectrum. Zeta-potential and dynamic light scattering (DLS) measurements were performed using a Zetasizer Nano ZS90 (Malvern Instruments, Malvern, UK). Transmission electron microscopy (TEM) images were observed using JEM-2200FS (Jeol Ltd., Tokyo, Japan) at an accelerating voltage of 200 kV. The sample was suspended in an aqueous solution under ultrasonic treatment before direct deposition on a copper grid and air-drying for TEM observation. The time decay curve of fluorescence life was measured using an FLS920 (Edinburgh Instruments, Livingston, UK) combined with a fluorescence-lifetime and steady-state spectrometer; each sample was repeated three times to obtain more reliable lifetimes.

### 4.3. Construction of the Assemblies

#### 4.3.1. Construction of Bi-Assemblies of EuW_10_/GL-22 and EuW_10_/Spm

Spermine (2.0 mM × 12.5 μL) and GL-22 were mixed with EuW_10_ (50 μM) in MES-NaOH buffer (10.0 mM, pH = 6.5), where the final volume of each one was 1.0 mL. The fluorescence spectrum was then recorded under the excitation of 500 nm after incubation at room temperature for 10 min. Based on the above procedures, the EuW_10_/GL-22 assembly was constructed.

Similarly, different amounts of Spm (2.0 mM, 0–125 μL) were added into MES-NaOH buffer (10.0 mM, pH = 6.5) containing 25 μL × 2.0 mM EuW_10_ under vigorous stirring. The final volume of each solution was fixed at 1.0 mL. After incubation at room temperature for 10 min, a fluorescence spectrum was recorded for each mixture under the excitation of 500 nm. After ratio optimization, the final concentrations of EuW_10_ and Spm were fixed at 50 μM and 50 μM, respectively.

#### 4.3.2. Construction of Tri-Assemblies of EuW_10_/Spm/GL-22 and EuW_10_/GL-22/Spm

A measure of 2.0 mM × 12.5 μL GL-22 was first introduced into MES-NaOH buffer (10.0 mM, pH = 6.5) containing 50 μM EuW_10_, then 2.0 mM Spm (0–125 μL) was introduced into the above mixture, keeping the total volume at 1.0 mL for the following experiment. The fluorescence spectrum was then recorded under an excitation of 500 nm after incubation at room temperature for 10 min.

In parallel, 2.0 mM GL-22 with a volume from 0.5 to 30 μL was added into the MES-NaOH buffer (10.0 mM, pH = 6.5) containing EuW_10_/Spm (50 μM/50 μM), where the final volume of each was 1.0 mL. The fluorescence spectrum was then recorded under the excitation of 500 nm after incubation at room temperature for 10 min. Therefore, the tri-assembly of EuW_10_/GL-22/Spm is constructed as above, but in a different adding sequence of the constituents.

### 4.4. Bacteria Culture

Gram-negative *E. coli* and Gram-positive *S. aureus* were cultured in Luria-Bertani (LB) agar at 37 °C. In addition, a bacterial suspension was obtained by inoculating an MRSA single colony into an LB medium cultured for 14–16 h at 37 °C and 180 rpm. Freshly harvested cells were used during each experiment. The quantification of the optical density at 600 nm (OD_600_) measurements was used to evaluate *E. coli* and *S. aureus* growth. The OD_600_ = 0.4–0.6 was used for subsequent experiments.

### 4.5. Anti-Bacteria Assay via Agar Plates

The individual EuW_10_ (2.5 μM), Spm (2.5 μM), and GL-22 (1.25 μM) were combined into binary (EuW_10_/Spm, EuW_10_/GL-22, GL-22/Spm) or ternary (EuW_10_/Spm/GL-22, EuW_10_/GL-22/Spm) assemblies in the phosphate buffer solution (PBS, negative control), and the test sample was mixed with new *E. coli* suspensions (OD_600_ = 0.2). After 3 h incubation at 37 °C, 0.2 mL of the diluted bacterial suspension was plated onto the LB agar, which was determined by a standard plate count method to count the cell viability of bacteria. The picture was taken, and colony counts were performed after 16–18 h of incubation at 37 °C.

### 4.6. Anti-Bacteria Assay Based on OD_600_

*E. coli* and *S. aureus* were sub-cultured into 20 mL LB broth and cultured at 37 °C with shaking to an OD_600_ of approximately 1.0 then diluted 103-fold using fresh sterile PBS to an optical density (OD_600_) of 0.3–0.35 and incubated for 3 h. Then, different concentrations of EuW_10_/GL-22/Spm (2:2:1) were introduced into *E. coli* or *S. aureus*, respectively. Finally, the cultures were grown for an additional 0–9 h, and 100 μL aliquots were taken for OD_600_ measurements using a Shimadzu UV-3600 spectrophotometer (Kyoto, Japan).

### 4.7. Reactive Oxygen Species (ROS) Assay

The *E. coli* were treated with the individual EuW_10_, Spm, and GL-22, either three individual constituents or their binary or ternary assembly. After incubation for 9 h, the *E. coli* was collected, and the intracellular ROS was measured using a standard assay. The bacterial samples were treated with 30 μL × 1.0 mM 2′,7′-dichlorofluorescein diacetate (DCFH-DA), a specific indicator for ROS generation to oxidize non-fluorescent DCFH-DA to fluorescent DFC. The fluorescence intensity of bacterial suspensions was registered with a Shimadzu RF-5301PC fluorescence spectrophotometer (λ_ex_ = 488 nm; λ_em_ = 525 nm).

### 4.8. Biofilm Elimination Assay

The capacity of bacteria to form biofilm was evaluated by crystal violet assay using 96-well microtiter plates. The diluted *E. coli* bacterial suspension (100 μL) of OD_600_ = 0.3–0.5 was then co-incubated with 900 μL of LB medium (pH = 6.0) for *E. coli* bacteria in 96-well plates. The samples were then continuously cultured for 8 h at 37 °C.

The *E. coli* bacteria were treated with PBS (control), EuW_10_, Spm, GL-22, Spm/GL-22, EuW_10_/GL-22, EuW_10_/Spm, EuW_10_/Spm/GL-22, EuW_10_/GL-22/Spm, Spm/GL-22/EuW_10_. After different disposals, bacterial biofilms were incubated at 37 °C for 8 h.

After the biofilm formation, the supernatant was removed and then sterilized water was used to wash the biofilm in each well to remove samples and dead bacteria.

Thereafter, 200 μL of 0.1% crystal violet staining solution was added to each well for 30 min of biofilm staining. Then, the PBS buffer was used to wash the biofilm in each well again, and the biofilms were then dried at room temperature. Finally, each well was blended with anhydrous alcohol/acetic acid (*v*/*v* = 1:1) to wash out the bacterial biofilm stuck on the hole wall. For the biofilm in each group, a multi-function microplate reader (Thermo Scientific, Varioskan LUK, Waltham, MA, USA) was used to monitor the OD_595_ value, and a digital camera was used to record the color.

## 5. Conclusions

In summary, based on the arginine-rich peptide of HPV E6, inorganic Eu-containing POM (EuW_10_) and a biogenic amine (Spm), we constructed two tri-assemblies with considerable luminescent and antibacterial enhancements over the constituents. The assembly features include enhanced membrane breaking and BME-adaptive ROS generation, which result in the promotion of antibacterial efficiency. The intrinsic mechanism of the enhanced antibacterial of the assemblies was essentially attributed to the peroxidase-like property of EuW_10_ and the endogenously active H_2_O_2_ in BME, which significantly promoted ROS generation in the physiological and pathophysiological processes of the antibacterial effect. Therefore, the involved H_2_O_2_ in bacteria and BME played vital roles in enhancing ROS generation, where the intelligent BME-responsive tungsten (W)-polyoxometalate cluster resolved the problem of severe biofilm infections and achieved unprecedented antibacterial efficacy. Therefore, the present study demonstrates a concept to construct unique bio-inorganic materials for antibacterial applications and will be helpful in the promotion of the development of antiviral agents via multi-assembly between a POM, biogenic amine, and peptide in the future.

## Data Availability

The data presented in this study are available on request from the corresponding author.

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
