# Peer review of "Adaptive Responses of a Peroxidase-like Polyoxometalate-Based Tri-Assembly to Bacterial Microenvironment (BME) Significantly Improved the Anti-Bacterial Effects"

_ijms, 2023, doi:10.3390/ijms24108858_

Round 1
Reviewer 1 Report
I have carefully evaluated the manuscript entitled " Adaptive Responses of a Peroxidase-like Polyoxometalate 2 Based Tri-assembly to Bacterial Microenvironment (BME) Significantly Improved the Anti-bacterial Effects". This manuscript presents the simple construction of tri-assemblies with considerable luminescent and antibacterial enhancements based on the arginine-rich peptide of HPV E6, inorganic Eu-containing 516 POM (EuW10) and a biogenic amine (Spm). The idea of the work is interesting. However, there is some points that should be addressed before publication. The abstract is not clear. Particularly the first sentence should be revised. The novelty of the article should be stated more precisely. Through reading the introduction, the significance of this research could not be caught. The characterization of materials by XRD, FTIR, and XPS is missing. It is recommended to include XRD, FTIR and XPS. Rather than a simple list of each experiment's results, the results and discussion section of this manuscript should reflect what representative conclusions can be drawn from these findings, what scientific problems are solved or addressed by the research, and what the significance of this research is to the research field. Compare the result with already existing similar materials. The prepared materials.
Author Response
Responses to Reviewer #1:
1) To present the novelty of this study more precisely, we have changed the first sentence of the abstract as: “The present study supplies a tertiary assembly between POM, peptide, and biogenic amine, which demonstrate a concept to construct a unique hybrid bio-inorganic materials for the antibacterial purpose and will be helpful to promote the development of antivirus agents in the future”. Besides, significant changes have also performed in the following parts of abstract for that (Please see the revised abstract).
2) The characterization of materials by NMR and FTIR spectra have been reported in a previous study of us (Colloids and Surfaces B: Biointerfaces 212 (2022) 112379), so we will not repeat them here in details.
3) The reported material has certain guiding significance for the abuse of antibiotics in clinical practice. This new material can quickly remove biofilm breeding bacteria and completely kill bacteria, and truly achieve the stage of economy, efficiency and visualization.
Besides, we have enriched the innovative points of this paper in the last section of introduction as: “Therefore, the present study provides a unique strategy to enhance the antibacterial effect of POMs through tripartite supra molecular coordination. It is straightforward, easy-operation, and cost-effective to fit both the biological and clinical needs. In addition, the detailed mechanism revealing both on the assembly construction and antibacterial pathway will help to expand the POMs application and further the development of super antibacterial materials in practice”. Please refer to lines 79 to 84 on page 2 for detailed description.

Reviewer 2 Report
The authors prepared small nanospheres assembly of EuW10, Spm and peptide GL-22. Furthermore, the intrinsic mechanism investigations revealed in detail that the encapsulation of EuW10 in Spm and further GL-22, enhanced the uptake abilities of EuW10 in bacteria, which further improved the ROS generation in BME via the abundant H2O2, and finally promoted the antibacterial effects significantly. The results are very interesting. However, some points of the manuscript should be improved. Specific comments are given below.
1. The authors should measure the stability of EuW10/Spm.
2. The sample of EuW10/Spm should be measured by XPS.
3. The authors should further discuss the effect of different amounts of GL-22 on the assembly of EuW10/Spm.
4. The novelty of this paper should be further specified in the introduction.
5. The molecular of chitosan should be offered.
6. The peroxidase-like activity should be measured.
7. Please carefully check the manuscript for writing and grammar.
Author Response
Responses to Reviewer #2:
1) The stability of the EuW10/Spm assembly is good. When the EuW10/Spm assembly was placed at room temperature for one-month, there is basically no fluorescence change and no particle size change.
2) The content of EuW10 in the assembly is low, so the signal of it in the XPS test is relatively weak. The signal-to-noise ratios are still poor after several times trying for detection. Sorry, we cannot supply it currently.
3) In the part of 2.1, we discussed in detail the influence of different concentrations of GL-22 on the assembly of EuW10/Spm. The luminescence intensity of the tri-assembly increased with the more GL-22 addition, while the particle size increased gradually (Fig. 3). Besides, during introducing of the positively charged GL-22, the negative charges exposed on the surface of EuW10 declined remarkably, as revealed by the Zeta-potential. Along with more GL-22 joining, the binding between EuW10 and GL-22 tends to be saturated (Fig.2C), and the particle size of the final assembly stop to increase.
4) Please see the answer to Q#3 of Reviewer#1.
5) We do not involve chitosan molecules throughout the whole experiment!
6) In Fig.7, we showed the peroxide-like activity of EuW10 and its assembly by fluorescence spectra.
7) We have carefully double-checked the spelling and grammar of the manuscript, thank you for the kind reminding.

Reviewer 3 Report
The manuscript by Wu et al. reports formation of POM, spermine and GL22 peptide associates and their utilization as biologically active materials. The presented data are in the scope of IJMS and can be published in the journal. However, some additional issues should be addressed before publication:
1) A general representation of the POM, spermine and GL22 should appear inside the Introduction part.
2) Formation of the tri-assembly of EuW10/Spm/GL-22 should be carefully studied. In my opinion, the detailed NMR experiments should appear in the MS.
3) What about stability of the above-mentioned tri-assembly? Some comments should appear in the main text. The same should be given for the stability of EuW10/Spm/GL-22 in PBS buffer.
4) What in the difference between EuW10/Spm/GL-22 and EuW10/GL-22/Spm (Part 3.3.). Some comments should be given.
5) Some TEM images of the corresponding biofilms should appear in the main text. It would be great if some elemental analysis of such objects will appear in the experimental section.
Author Response
Responses to Reviewer #3:
1) The general introduction of the POM, spermine and GL-22 were included in the introduction section. For details, see lines 57 to 60 on page 2, and their structure diagram are new added in the Supporting Information, as Figure S1D.
2) The characterization of materials by NMR and FTIR have been described in a previous report of us, please see Colloids and Surfaces B: Biointerfaces 212 (2022), 112379, for details.
3) The stability comments of both tri-assemblies were added as: “Based on the emission intensity check both the tri-assemblies were stably enough for 3-month preservation in buffer solution, and their antibacterial properties were conserved well.”, please refer to lines 215 to 217 on page 7 of the revised manuscript.
4) The two assemblies combine differently, as shown in the newly added Scheme 1: the final fluorescence enhancement of the two assemblies was obviously different, which was described in line 115 to 119 on page 3 of the manuscript. In the biofilm elimination experiment, the ability of the two assemblies to destroy the biofilm was also different.
Scheme 1. Step-up fluorescence enhancements of the assemblies and the improved antibacterial activity of them.
5) The biofilms that can be imaged by staining under TEM generally require 1 to 2 months of bacterial incubation and culture; while the crystal violet staining usually only requires 1 day of bacterial incubation to obtain thin layers of biofilms. In order to obtain the experimental results intuitively, simply and quickly, we do not recommend using TEM to observe the biofilms, as which is too much rely on the thickness of biofilms and staining techniques. In addition, we conducted energy dispersive spectrum tests for the tri-assembly in Figure 3D, and the corresponding results were added in the supporting information as Figure S2.

Round 2
Reviewer 1 Report
The revised manuscript is recommended for publication.
Author Response
Thank you for your recommendation!

Reviewer 2 Report
The authors have revised the manuscript. However, some points of the manuscript should be improved. Specific comments are given below.
1. The authors should measure the stability of EuW10/Spm. The stability should be measured by DLS.
2. The peroxidase-like activity should be measured. TMB, ABTS and OPD should be used as substrates to evaluate the peroxidase-like activity.
Author Response
Responses to Reviewer #2:
As you suggested, we have added the related experiments, DLS of Figure S3 (to assay the stability of EuW10/Spm), and enzymatic assay of Figure S5 (using TMB as substrate to evaluate the peroxidase-like activity of EuW10; and EuW10/Spm; EuW10/Spm/GL-22), respectively, in the supporting information.
Meanwhile the description for Figure S3 was added as: “As assayed by DLS (Figure S3), after 30-day incubation in the buffer solution no much change for the particle size of assembly was observed, which validate the excellent stability of it.” in page3 of the main text.
Figure S3. Time-dependent (A) size distributions and (B) histogram of EuW10/Spm (50 μM/50 μM) in 30 days.
While that for Figure S5 was added as: “the peroxidase-like activity of EuW10 and assemblies were measured using TMB as a substrate (Figure S5). Further experiments show the assemblies promotes the TMB oxidation significantly. For example, in the presence of EuW10, the absorbance of oxTMB changed weakly (Figure S5A); however, the EuW10/Spm increased the characteristic absorption of A652 by 14.6-fold within 5 min (Figure S5B). At the identical conditions, the oxidation rate of TMB is significantly increased by EuW10/Spm/GL-22 as A652 increases faster than EuW10 and EuW10/Spm (Figure S5C).” in page 9 of the main text.
As the production of ROS produced by EuW10 and its assembly were either assessed using a DCFH-DA method (Please see Figure S6), so we did not perform the data using other substrates as ABTS and OPD.
Figure S5. Time-dependent UV-vis absorption spectra of TMB (0.5 mM) in the presence of (A) EuW10 (50 μM); (B) EuW10/Spm (50 μM/50 μM); (C) EuW10/Spm/GL-22 (50 μM/50 μM/35 μM). (D) The plots of corresponding intensity changes at 652 nm for (A), (B) and (C).

Reviewer 3 Report
The MS can be accepted for publication at the current stage.
Author Response
Thank you for your suggestions.

Round 3
Reviewer 2 Report
The authors have addressed the problem very well, and the manuscript can be accepted in the present form.